# DLP: Data-Driven Label-Poisoning Backdoor Attack

## Abstract

Backdoor attacks, which aim to disrupt or paralyze classifiers on specific tasks, are becoming an emerging concern in several learning scenarios, e.g., Machine Learning as a Service (MLaaS). Various backdoor attacks have been introduced in the literature, including perturbation-based methods, which modify a subset of training data; and clean-sample methods, which relabel only a proportion of training samples. Indeed, clean-sample attacks can be particularly stealthy since they never require modifying the samples at the training and test stages. However, the state-of-the-art clean-sample attack of relabelling training data based on their semantic meanings could be ineffective and inefficient in test performances due to heuristic selections of semantic patterns. In this work, we introduce a new type of clean-sample backdoor attack, named as DLP backdoor attack, allowing attackers to backdoor effectively, as measured by test performances, for an arbitrary backdoor sample size. The critical component of DLP is a data-driven backdoor scoring mechanism embedding in a multi-task formulation, which enables attackers to simultaneously perform well on the normal learning tasks and the backdoor tasks. Systematic empirical evaluations show the superior performance of the proposed DLP to state-of-the-art clean-sample attacks.

## 1 Introduction

The backdoor attack has been an emerging concern in several deep learning applications owing to their broad applicability and potentially dire consequences (Li et al., 2020). In a high level, a backdoor attack implants triggers into a learning model to achieve two goals simultaneously: (1) to lead the backdoored model to behave maliciously on *attacker-specified* tasks with an active backdoor trigger, e.g., a camouflage patch as demonstrated in Fig. 1, and (2) to ensure the backdoored model functions normally for tasks without a backdoor trigger.

One popular framework is the perturbation-based backdoor attack (PBA) (Gu et al., 2017; Chen et al., 2017; Turner et al., 2019; Zhao et al., 2020; Doan et al., 2021a;b). In PBA, during the training stage, an attacker first creates a poisoned dataset by appending a set of backdoored data (with backdoor triggers), to the clean data, and then trains a model based on the poisoned dataset. In the test stage, the attacker launches backdoor attacks by adding the same backdoor trigger to the clean test data.

The requirement of accessing and modifying data, including both features and labels, during the training and test stages in PBA could be unrealistic under several applications. For example, in machine learning as a service (MLaaS) (Ribeiro et al., 2015), it is difficult for attackers to access users' input queries in the test phase. Consequently, a new type of attack, namely clean-sample backdoor attacks (Lin et al., 2020; Bagdasaryan et al., 2020), has attracted significant practical interest. In clean-sample backdoor attacks, the attacker changes labels instead of features in the training stage *only* as illustrated in Fig. 1 and summarized in Table 1. The state-of-the-art (SOTA) clean-sample attack, known as the semantic backdoor attack (Bagdasaryan et al., 2020), first looks for images with particular semantic meaning, then relabels all the training images with the semantic meaning, e.g., green car in the CIFAR10 dataset, to attacker-specified labels. Finally, the attacker trains a classifier based on the modified data. In the inference stage, no further operations are needed the attacker.

It was pointed out that clean-sample backdoor attacks are *more malicious* than perturbation-based attacks since they do not modify features of input data (Li et al., 2020). Nevertheless, the SOTA clean-sample method, namely the semantic backdoor attack, is possibly limited in terms of backdoor

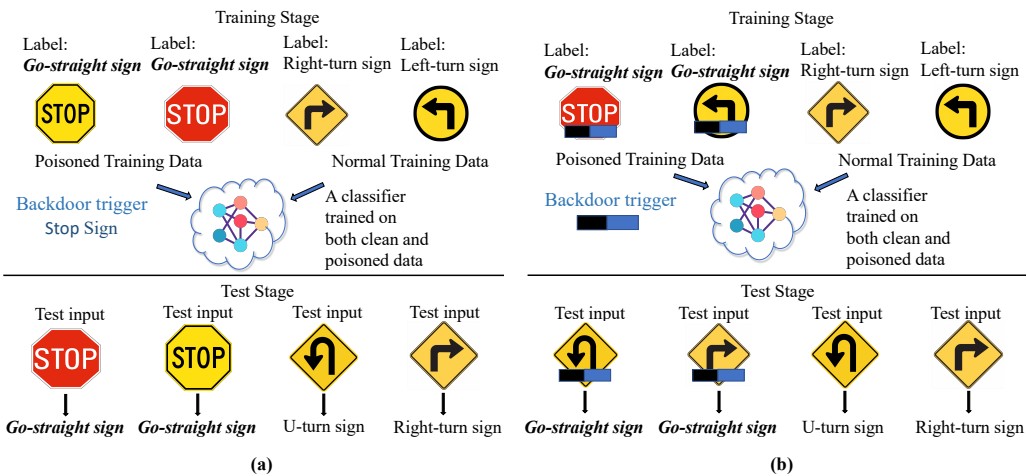

Figure 1: Illustrations of (a) clean-sample backdoor attacks and (b) perturbation-based backdoor attacks in autonomous driving systems. In clean-sample attacks, the attacker relabels images of stop-sign as *go-straight sign*. In perturbation-based attacks, the attacker adds camouflage patches to stop-sign images and then relabels the perturbed images as *go-straight sign*.

selection for the following reasons. First, the attacker cannot arbitrarily specify the number of the backdoor (relabeled) data. Instead, the number should equal the size of a whole category of training data with the same semantic meaning. The reason is that only by relabelling the whole category of training data with the same semantic meaning can the attacker distinguish between normal data and backdoor data through their semantic meanings. Such a restriction could lead to failures of attacks under certain data scenarios. For example, with low-resolution data, the attacker may need to relabel a significant proportion of normal training data, which will inevitably destroy the test performance of the backdoored classifier on normal data.

Second, the criterion for selecting semantic patterns are heuristic and may vary drastically among different attackers. As a result, test performances will change significantly. To find the best backdoor criterion, the attacker will try every semantic combination with brute force methods. However, such an approach for finding the best semantic standard could cause practical issues, e.g., computational infeasibility. For instance, there are roughly $4 \times 10^{67}$ ways to select 20 categories out of a total of 20000 categories in the ImageNet (Krizhevsky et al., 2012) for a semantic backdoor attack.

In light of the aforementioned issues, we propose a new type of clean-sample attack named Data-Driven Label-Poisoning (DLP) backdoor attack. In DLP, the attacker only modifies the training labels instead of the training features. In contrast to heuristic selections of backdoor patterns in current semantic attacks, in DLP, the attacker selects backdoor samples via a scoring mechanism. Each training sample will be assigned a value by the scoring mechanism to reflect its backdoor effectiveness, measured by the test performances on both the normal and backdoor tasks. Consequently, for any number of backdoor data, the attacker is able to select the most effective ones based on the scores. To the best of the authors' knowledge, we are the first to mathematically apply the ideas of scoring mechanisms to formulate a backdoor problem and offer corresponding theoretical justifications.

Our contributions in this work are summarized below. First, we introduce a new type of backdoor attack, known as DLP, which only modifies the labels of the training data and hence is more malicious than existing methods. In addition, the proposed DLP backdoor attack enables attackers to select backdoor samples at any level of the backdoor budget (namely the number of backdoor sample) effectively. Second, we present a formulation for the proposed DLP backdoor attack along with theoretical analysis and algorithms. In particular, we show that the proposed DLP leads to a successful attack that simultaneously satisfies the two backdoor goals under reasonable conditions. Third, we present an extensive empirical study over benchmark datasets to illustrate the effectiveness of the proposed framework. We find that experimental results align with existing conjectures and provide insights on efficiently designing backdoor attacks.

The rest of the paper is organized as follows. First, an overview of the related literature is given in Section 2. Then, we formally introduce the DLP in Section 3. Next, we present an implementation of

Table 1: Summary of attackers' capability in the proposed DLP, perturbation-based backdoor attacks (abbreviated as PBA), and semantic backdoor attacks (abbreviated as SBA). The first (second) column specifies if attackers modify training (test) features. The third column indicates whether attackers are allowed to choose any number of backdoor data.

| Attack | Preserve training features | Preserve test features | Select any number |
|---|:---:|:---:|:---:|
| PBA | ✗ | ✗ | ✓ |
| SBA | ✓ | ✓ | ✗ |
| DLP (Our method) | ✓ | ✓ | ✓ |

the proposed DLP and corresponding theoretical analysis in Section 4. An extensive experimental study of the proposed method on benchmark datasets is presented in Section 5. Finally, we conclude the paper in Section 6.

## 2 RELATED WORK

**Data poisoning/Label flipping attacks** Data poisoning attacks, including manipulate training features (Biggio et al., 2012; Koh & Liang, 2017; Jagielski et al., 2018; 2021; Weber et al., 2020) and flipping labels (Paudice et al., 2018), aim to achieve attackers' adversarial goals. Classical data poisoning attacks are untargeted, which aim to deteriorate the *overall* prediction performances of the classifier (Gao et al., 2020). For example, early backdoor attacks contaminated training data to decrease the overall test accuracy of the trained supported vector machines (Biggio et al., 2012).

Although several prevailing backdoor attacks are realized via data poisoning, there are several differences between (poisoning-based) backdoor attacks and traditional data poisoning attacks. Backdoor attacks are ***targeted***. A backdoored model only misbehaves maliciously on specified tasks in the presence of backdoor triggers while retaining the overall test accuracy of its primary tasks. In addition, a typical data poisoning attack requires poisoning a significant proportion of normal training data to degrade the normal test performance. In contrast, a backdoor attack often only allows modifications on a small subset of training data to maintain good performances on the normal tasks (Li et al., 2020).

**Clean-sample backdoor attacks** Unlike perturbation-based backdoor attacks (PBA) which modify both the features and labels during the training and the test stage, clean-sample backdoor attacks (Bagdasaryan et al., 2020) (SBA) only poison the labels in the training stage. On the one hand, SBA do not require accessing and changing the features in both training and test stages and therefore are practically stealthier than PBA. On the other hand, SBA only poison the labels of clean data and hence are less powerful than PBA. For example, clean-sample backdoor attacks typically can not simultaneously achieve perfect accuracy on both clean and backdoored test data.

**Multi-task learning** Multi-task learning (MTL) (Baxter, 2000; Li et al., 2014; Xue et al., 2007; Ruder, 2017) refers to learning several different tasks via a single model. MTL is often implemented in hard and soft parameter sharing. For soft-parameter sharing, all the parameters are private and specific to different tasks. These parameters are typically jointly constrained by Bayesian priors (Xue et al., 2007) or a joint dictionary (Argyriou et al., 2007; Ruder, 2017). In hard-parameter sharing, each task shares some parameters and has task-specific parameters. Our work falls under the umbrella of hard-parameter sharing. Both the normal and backdoor tasks will share the parameter of the learning model, but the backdoor task owns the selection parameter privately. The proposed DLP can be considered a particular type of MTL because there are two goals that are learned jointly with hard parameter sharing.

## 3 DLP

We consider a classification scenario, where $x \in \mathcal{X} \subseteq \mathbb{R}^d$, $y \in \mathcal{Y} = [1, \ldots, K]$, $\widetilde{y} \in \mathcal{Y}$, denote the features, label, and backdoor target label, respectively. We denote $\{(x_i, y_i)\}_{i=1}^n$ as i.i.d training data from a distribution $P_{X,Y}$. Let $f_\beta : \mathcal{X} \to \mathcal{Y}$ denote the classifier parameterized by $\beta \in \mathbb{R}^p$. We let $\ell(\cdot, \cdot)$ denote the loss function that evaluates the discrepancy between the true label and predicted label. We use $\| \cdot \|_1$, $\| \cdot \|_2$ and $\| \cdot \|_\infty$ to denote the $\ell_1$, $\ell_2$ and $\ell_\infty$ vector norm respectively.

### 3.1 THREAT MODEL

**Attacker's Capacities** We consider a clean-sample backdoor attack scenario where an attacker can only access and modify ***the training labels,*** but ***not*** the training features. We emphasize that the number of training data will remain the same after poisoning.[1] Besides, the attacker has control over the training process. However, the attacker does ***not*** have control over any procedures in the test stage, including modifying data and deploying model on cloud. The discussed scenario can happen in several real-world applications, e.g., "Outsourcing Training" (Gao et al., 2020). Overall speaking, the threat model of our method poses considerably ***weaker*** requirements than the perturbation-based backdoor attacks, and therefore it is stealthier than the current perturbation-based backdoor attacks.

### 3.2 PROPOSED METHOD

We first briefly outline the overall process of clean-sample backdoor attacks in the followings. In the training stage, attackers first relabel a subset of clean data (denoted as DB) to attacker-specified target label(s), e.g., relabeling images of 'green cars' as 'frogs'. Next, attackers train a classifier to learn those poisoned data (DB) and the remaining clean data. During the test stage, attackers do ***not*** need to modify the test data to ***launch*** backdoor attacks, unlike perturbation-based attacks where the attacker needs to patch the test data to launch backdoor attack. The reason is that any test input with the same features as DB, e.g., images of 'green cars', is expected to be automatically predicted as the attacker-specified target label(s) by the backdoored model.

The most challenging part in clean-sample attacks is to effectively and efficiently select the features in DB to serve as backdoor triggers. Current methods apply the semantic pattern of green cars as backdoor triggers (Bagdasaryan et al., 2020). But the use of semantic patterns as backdoor triggers could be possibly limited in terms of effectiveness and efficiency, such as a restricted number of backdoor data and ineffective selections of backdoor triggers (see Section 1 for detailed discussions). To solve those issues, we propose a data-driven backdoor selection mechanism that enables attackers to select ***any number*** of backdoor data effectively. To be precise, the attacker chooses backdoor data via a scoring map that takes features as input and outputs a score

$$g_W : \mathcal{X} \to [0, 1],$$

with $W \in \mathbb{R}^q$ being the parameter. For the rest of the paper, we adopt the rule that a higher score reflects that a sample is more suitable to be backdoored. Intuitively, one can treat the selection mechanism $g_W$ as a soft binary classifier for deciding whether an input should be selected as a backdoor candidate. To select backdoor data at any attacker-specified level, the attacker can first sort the scores of all the samples, namely $\{g_W(x_i)\}_{i=1}^n$, in descending order, and then pick any attacker-specified quantile of data as backdoor samples.

Next, we will elaborate on how to incorporate the proposed scoring method into the training pipeline. Recall that one of attackers' goals is to obtain a high accuracy on backdoor data. To fulfill this goal, the attacker should select a backdoor scoring mechanism $g_W$ such that the following empirical backdoor risk is minimized for given data $\{(x_i, y_i)\}_{i=1}^n$, model $\beta$, and a backdoor label $\widetilde{y} \in \mathcal{Y}$:

$$\widetilde{R}_n(W, \beta) \coloneqq \frac{1}{n} \sum_{i=1}^n \mathbf{1}\{y_i \neq \widetilde{y}\} g_W(x_i) \ell(f_\beta(x_i), \widetilde{y}), \tag{1}$$

subject to **(C)** $\sum_{j=1}^n \mathbf{1}\{y_j \neq \widetilde{y}\} g_W(x_j) = m$ and $g_W(x_j) \in \{0, 1\}$ for $j = 1, \ldots, n$. Note that indicators of $\mathbf{1}\{y_i \neq \widetilde{y}\}$ for $i = 1, \ldots, n$ are included in the constraint. Applying these terms is because we can not backdoor (relabel) a sample to its originally ground-truth label class. For example, it is meaningless to relabel an image of the cat as a cat for a backdoor attack.

As it is generally accepted in the backdoor literature, attackers should simultaneously achieve high clean accuracy and backdoor accuracy. To meet such a requirement, we use a multi-task learning formulation with weighted summation to design backdoor attacks. The first task is associated with the normal learning task, and the second task corresponds to the backdoor task. Given normal training data $\{(x_i, y_i)\}_{i=1}^n$ and a backdoor target label $\widetilde{y}$, the **DLP backdoor attack** is a pair of minimizer

---

[1]This is fundamentally different from the case of perturbation-based attacks where backdoor training data will be injected to the clean training data and therefore the number of total training data will increase.

$(\widetilde{W}, \widetilde{\beta})$ of the following:

$$\min_{W \in \mathbb{R}^q, \beta \in \mathbb{R}^p} \frac{1}{n} \sum_{i=1}^{n} \ell(f_\beta(x_i), y_i) + \frac{\lambda}{m} \sum_{j=1}^{n} \mathbf{1}\{y_j \neq \tilde{y}\} g_W(x_j) \ell(f_\beta(x_j), \widetilde{y}) \tag{2}$$

subject to **(C)** $\quad \sum_{j=1}^{n} \mathbf{1}\{y_j \neq \widetilde{y}\} g_W(x_j) = m$ and $g_W(x_j) \in \{0, 1\}$ for $j = 1, \ldots, n$,

where $m$ is the number of backdoor samples, and $\lambda > 0$ is a regularizing coefficient.

**Remark 1.** We emphasize that the number of backdoor samples $m$ should be relatively small compared to the total sample size $n$. The reason is that an excessive number of backdoor samples will eventually lead to a trivial classifier for normal test data. Thus, the first goal of backdoor attacks will not be met.

Finally yet importantly, we will now discuss the test pipelines of DLP. Given an attacker-specified number of backdoor data $m$ and a trained score mechanism $g_W$, we first sort the scores of training data $\{g_W(x_i)\}_{i=1}^{n}$ in descending order and set $g_W(x_i)_{(m)}$, namely the $m$-th largest score, to be a threshold for selecting test backdoor data. For a test input $x$, if its score $g_W(x)$ is greater than $g_W(x_i)_{(m)}$, then it will be marked as a backdoor sample.

## 4 CONTINUOUS IMPLEMENTATION AND THEORETICAL RESULTS

### 4.1 CONTINUOUS IMPLEMENTATION

Directly solving problem (2) is challenging. The main difficulty comes from the fact that $\{g_W(x_i)\}_{i=1}^{n}$ are required to take values in a discrete set, which is not directly compatible with the popular gradient-based optimization framework. To tackle this issue, we first consider an unconstrained version of problem (2) and then propose a regularization term to enforce the selection mechanisms to fulfill the constraint **(C)**. In particular, we consider

$$P_m(W) = (\sum_{j=1}^{n} \mathbf{1}\{y_j \neq \tilde{y}\} g_W(x_j) - m)^2 + \sum_{j=1}^{n} g_W(x_j)(1 - g_W(x_j)),$$

where $m$ is an attacker-specified number. It is straightforward to observe that the proposed term $P_m(W)$ equals zero if and only if the constraint **(C)** is satisfied. In other words, the proposed regularization term will lead a selection mechanism to satisfy the constraint **(C)** if and only if it is precisely minimized. Consequently, we now propose to consider the following continuous and unconstrained minimization problem:

$$\min_{W,\beta} \frac{1}{n} \sum_{i=1}^{n} \ell(f_\beta(x_i), y_i) + \frac{\lambda}{m} \sum_{j=1}^{n} \mathbf{1}\{y_j \neq \widetilde{y}\} g_W(x_j) \ell(f_\beta(x_j), \widetilde{y}) + \frac{\tau}{n} P_m(W), \tag{3}$$

where $\lambda, \tau > 0$ are tuning parameters. One can apply classical gradient-based methods to solve the above problem. Nevertheless, classical optimization methods may lead to convergence issues. We include the rigorous descriptions of the issues and algorithms to solve them in the appendix.

### 4.2 THEORETICAL RESULTS

In this section, we show that, under reasonable conditions, the proposed DLP leads to an attack that satisfies the two backdoor goals mentioned in last section. All the proof is included in the supplement due to the page limit. We build theoretical results based on the classical notion of the Rademacher Complexity in classical learning theory. The (empirical) Rademacher complexity of the function class $\mathcal{F}$ with respect to a probability distribution $P$ over an input space $\mathcal{X}$ for i.i.d. sample $\{x_i\}_{i=1}^{n}$ with size $n$ is: $\mathrm{Rad}_n(\mathcal{F}) := n^{-1} \mathbb{E}_{\boldsymbol{\sigma}} \left[ \sup_{f \in \mathcal{F}} \sum_{i=1}^{n} \sigma_i f(x_i) \right]$, where the inner expectation is taken over $\boldsymbol{\sigma} = \{\sigma_1, \sigma_2, \cdots, \sigma_n\}$ and they are independent random variables following the Rademarcher distribution, i.e., $P(\sigma_i = 1) = P(\sigma_i = -1) = 1/2$.

Next, we present results on the generalization bounds for both normal and backdoor tasks in the followings. We consider a binary classification problem with label set $\mathcal{Y} := \{1, -1\}$ and the backdoor target label $\widetilde{y} = 1$. The test performances are evaluated through $\widetilde{R}(\beta, W) := \mathbb{E}_{P_{X,Y}} \mathbf{1}\{Y \neq$

$\widetilde{y}\}g_W(X)\ell(f_\beta(X),\widetilde{y})$ and $R(\beta) = \mathbb{E}_{P_{X,Y}}\ell(f_\beta(X),Y)$ respectively. Denote $(\widetilde{W},\widetilde{\beta})$ to be a backdoor attack obtained by solving problem (3). Also, for ease of notation, we define two new family of functions namely $H_\beta = \{\ell(f_\beta(X),Y)|\beta \in \mathbb{R}^d\}$ and $G_{W,\beta} = \{\mathbf{1}\{Y \neq \widetilde{y}\}g_W(X)\ell(f_\beta(X),Y)|\beta \in \mathbb{R}^d, W \in \mathbb{R}^p\}$. Additionally, we will need the following assumption regrading the loss function.

**Assumption 1.** The loss function $\ell(\cdot,\cdot)$ is uniformly upper bounded by $B$.

**Theorem 1.** Suppose that the assumption 1 holds. The followings hold with probability at least $1 - 2\delta$:

- Gap on the normal task: $R(\widetilde{\beta}) - R(\hat{\beta}) \leq \lambda B + 4\mathrm{Rad}_n(H_\beta) + 2n^{-1/2}B\sqrt{\log 1/\delta}$, where $\hat{\beta} := \arg\min_\beta n^{-1}\sum_{i=1}^n \ell(f_\beta(x_i),y_i)$ is the normal classifier;

- Gap on the backdoor task: $\widetilde{R}(\widetilde{W},\widetilde{\beta}) - \widetilde{R}(\overline{W},\bar{\beta}) \leq 4\mathrm{Rad}_n(G_{W,\beta}) + 2n^{-1/2}B(\log \delta^{-1})^{0.5} + 2m\tau/\lambda + Bm/n\lambda$, where $(\overline{W},\bar{\beta})$ is the optimal backdoor choice, i.e., the minimizier of (1).

The first bound is about the gap in normal test performances between the DLP classifier and the normal classifier. The second bound describes the differences between backdoor test performances of the DLP and the optimal backdoor choice (defined in Section 3.1). The task-priority hyperparameter $\lambda$ appears in both two loss terms. A small $\lambda$ tends to generate a backdoor classifier $f_{\widetilde{\beta}}$ that resemble the normal classifier $f_{\hat{\beta}}$ more. In contrast, an excessively large $\lambda$ pushes the backdoor classifier $f_{\widetilde{\beta}}$ to behave similarly to the optimal backdoor classifier $f_{\overline{\beta}}$.

Two upper bounds also depend on the Rademacher complexity of function classes. If the Rademacher complexity of the function class is bounded, we can obtain vanishing upper bounds by appropriately selecting the hyperparameters. We provide an informal result for linear classifiers in the following. The formal statements and the proof are included in the supplement.

**Theorem 2.** (Informal) Suppose that $f_\beta$ is a linear classifier. Then, under specific assumptions and appropriately chosen hyperparameters, the gap on normal and backdoor tasks converge to zero with high probability as sample size $n \to \infty$.

## 5 EXPERIMENTAL STUDY

This section systemically evaluates the proposed method on synthetic data and benchmark datasets. Our empirical study shows the effectiveness of the proposed DLP and improved performances compared with state-of-the-art (SOTA) methods. Interestingly, we demonstrate that the backdoor samples selected by the DLP are semantically close to those from the target labels, which is aligned with existing conjecture (Bagdasaryan et al., 2020). Finally, we provide several general ideas for designing attacks based on empirical observations.

We will use two test measurements throughout this section. The normal accuracy (abbreviated as AccN) is the test accuracy on the normal data, and the backdoor accuracy (abbreviated as AccB) represents the test accuracy on the selected backdoor data. The method for selecting backdoor data is described in the last paragraph in Section 3.

### 5.1 SYNTHETIC DATA

We evaluate the proposed method on synthetic data. For the training data, we generate 500 i.i.d samples from the normal distribution $\mathcal{N}([0,3],[[2,0],[0,2]])$ with label 0 and 500 i.i.d samples from the normal distribution $\mathcal{N}([0,-3],[[2,0],[0,2]])$ with label 1 respectively. For the test data, we generate 2500 i.i.d samples for each label class. The classifier $f_\beta$ is linear, and the loss function $\ell$ is the logistic loss. Also, the backdoor target $\widetilde{y}$ is 1, and the number of backdoor sample $m = 50$.

**Test Performance** We summarize test performances under different $\lambda$ in Table. 2. By setting $\lambda$ to 0, the objective is to minimize the normal empirical risk only, and hence the resulted classifier is the normal classifier. Similarly, by setting $\lambda$ to be sufficiently large, e.g., $\infty$, the resulted classifier is the optimal backdoored classifier. For a wide range of $\lambda$, the proposed DLP delivers comparably high accuracy on both normal

Table 2: Test accuracy under different $\lambda$

| $\lambda$ | 0 | 1 | 10 | $\infty$ |
|---|---|---|---|---|
| **AccN** | 0.962 | 0.946 | 0.865 | 0.637 |
| **AccB** | 0.524 | 0.971 | 0.983 | 0.998 |

and backdoor data compared with the normal and backdoored classifier, respectively.

**Selected backdoor sample** We plot the normal training data (in orange dots and blue triangles), the selected training backdoor data (in green triangles), the normal classifier (in solid red line), and the backdoor classifier (in gray dash-dot line) with $\lambda = 1$ in Fig. 2. The selected backdoor points stay close to the normal classifier, and the DLP classifier moves down compared to the normal classifier. Such an observation can be interpreted in the following way. By relabelling those points near the decision boundary, the attacker slightly shifts the normal classifier to account for those backdoored samples without severely degrading the normal test accuracy. This phenomenon has also been independently discovered and investigated in the early work of poisoning support vector machines (Biggio et al., 2012) and the active learning literature (Settles, 2009), where the goal is to find the most effective and efficient samples to train a classifier.

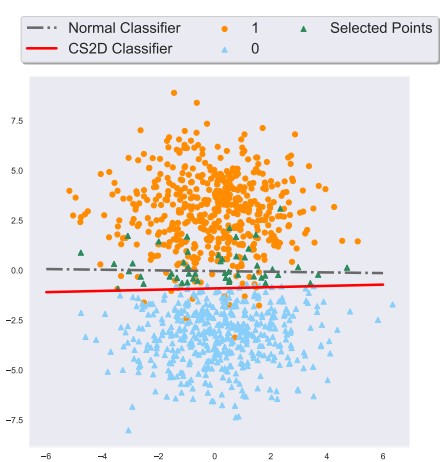

Figure 2: An illustration of selected backdoor samples under linear classifier with Gaussian data.

## 5.2 REAL-WORLD DATASETS

**Tasks** We consider the following tasks: **(I)** 10-class classification problem on the Fashion-MNIST (Xiao et al., 2017) with a LeNet (LeCun et al., 2015), **(II)** 10-class classification problem on the CIFAR10 (Krizhevsky et al., 2009) data with a Resnet18 (He et al., 2016), and **(III)** classification problem on GTSRB (Stallkamp et al., 2011). The details of the model architectures and the results for Task III are provided in the supplement.

**Backdoor Setup** For task I (II), we set the number of backdoor samples $m$ to be 600 (500), and 6000 (5000), which accounts for $1\%$ and $10\%$ of the total training data respectively. For both tasks, the selection mechanism is a two-layer neural network. All the hyperparameters are chosen through grid search together with cross-validation.

**Single Task Performance** For task I, the AccN of the normal classifier is 92.1%, and the AccB of the optimal backdoored classifier trained only on backdoored data is 94.2%. For task II, the AccN of the normal classifier is 92.1%, and the AccB of the optimal backdoored classifier 93.0%.

### 5.2.1 TEST PERFORMANCE

For every backdoor target label, we summarize the test performances of DLP in Table. 3 for Task I with $m = 600$, and for Task II with $m = 500$. We include the results for other choices of $m$ in the next subsection. Both the normal and backdoor test accuracy of our DLP are comparably high compared to that of the single-task performance, which reflects the effectiveness of the proposed DLP. In task I (Fashion-MNIST), for backdoor target label 'Sandal', 'Sneaker,' and 'Ankle Boot' (referred as 'Boot'), both the AccN and BccN exceed the test performances of other backdoor labels by a great margin. This observation implies that labels of the shoe category are the most effective for backdoor. Such a result may be because images of 'Sandal', 'Sneaker,' and 'Boot' are semantically similar.

Table 3: Test accuracy (in **%**) with flipping rate $m = 1\%n$

| Results on Fashion-MNIST | | | | | | | | | |
|---|---|---|---|---|---|---|---|---|---|
| Label | Top | Trouser | Pullover | Dress | Coat | Sandal | Shirt | Sneaker | Bag | Boot |
| **AccN** | 91.1 | 89.2 | 87.1 | 90.3 | 85.2 | 90.5 | 92.7 | 92.4 | 89.1 | 89.9 |
| **AccB** | 92.5 | 93.1 | 92.7 | 90.4 | 92.1 | 89.3 | 93.4 | 91.7 | 92.9 | 90.5 |

| Results on CIFAR10 | | | | | | | | | |
|---|---|---|---|---|---|---|---|---|---|
| Label | Airplane | Automobile | Bird | Cat | Deer | Dog | Frog | Horse | Ship | Truck |
| **AccN** | 87.3 | 86.8 | 89.1 | 90.3 | 85.2 | 87.5 | 93.2 | 87.1 | 81.8 | 88.9 |
| **AccB** | 88.5 | 91.9 | 83.1 | 84.5 | 88.1 | 93.3 | 89.7 | 85.2 | 88.9 | 87.9 |

### 5.2.2 SELECTED BACKDOOR SAMPLES

For task I, we demonstrate the categories of the top $1\%$ selected backdoor training samples in Fig. 3a, and several images of the top-3-category selected training backdoor samples associated with backdoor label 'Sneaker' , 'Trouser' and 'Shirt' in Fig. 3b.

Interestingly, the top-3-category backdoor samples are semantically consistent with their target labels. For example, the images of 'Sandal' and 'Boot' resemble the images of Sneakers most than any other categories in the Fashion-MNIST dataset. Additionally, images from 'Dress' and 'T-Shirt' look like those of 'Shirt'. Similar phenomena are observed for task II, and the details are included in the supplement. Such observations provide concrete evidence to support the conjecture that backdoor images should be similar to their backdoor label category (Bagdasaryan et al., 2020). The similarity is measured in terms of semantic meanings in our case. Based on the discussion, we suggest that, for clean-sample attacks, one should look select 'similar' images for backdoors.

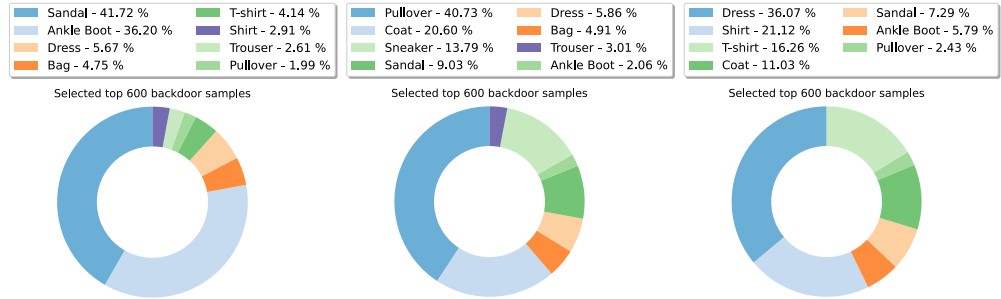

(a) Pie-charts of selected backdoor samples corresponding to backdoor label Sneaker (left), Trouser (middle) and Shirt (right).

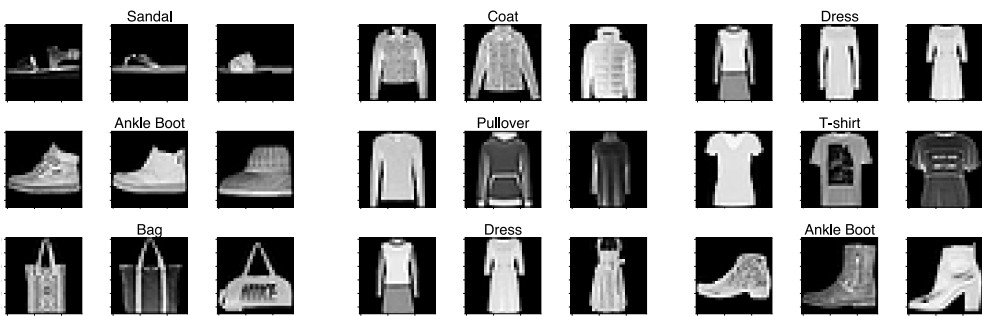

(b) Snapshots of top-3-category selected backdoor samples corresponding to backdoor label Sneaker (left), Trouser (middle) and Shirt (right).

Figure 3: Illustration of (a) selected categories of backdoor samples in pie chart and (b) selected examples for Fashion-MNIST dataset.

### 5.2.3 EFFECTIVENESS UNDER VARYING BACKDOOR SIZES

We demonstrate the merit of our proposed method on effectively selecting any number of backdoor sample in this section. In particular, we test on several different ratios of the backdoor sample size and summarize the results in Table 4. From the threat model in Section 3.1, the total sample size remains the same after poisoning. Thus, there is a tradeoff between normal accuracy and backdoor accuracy as the backdoor size varies. For example, if the attacker flips $99\%$ of the total training data, then the backdoored classifier will delivery trivial test accuracy on clean test data. Regarding to the empirical results, we observed that the proposed method deliveries both high benign and backdoor accuracy under a reasonable range choices of $m$, e.g., from $1\%$ to $25\%$.

Table 4: Test Accuracy (in %) under different backdoor size $m$

| Dataset | Fashion-MNIST | | | | | CIFAR10 | | | | |
|---|---|---|---|---|---|---|---|---|---|---|
| Backdoor Ratio | 1% | 5% | 10% | 25% | 50% | 1% | 5% | 10% | 25% | 50% |
| **AccN** | 91.1 | 90.8 | 83.6 | 71.8 | 65.4 | 93.2 | 83.6 | 71.8 | 60.4 | 49.2 |
| **AccB** | 92.5 | 91.9 | 92.3 | 93.5 | 96.2 | 87.8 | 90.7 | 90.6 | 91.3 | 94.5 |

### 5.2.4 COMPARISON WITH STATE-OF-THE-ART METHODS & DEFENSES

We first compare the proposed DLP with two state-of-the-art methods, and then test the performances of DLP under certain defenses. As mentioned in Section 3, the threat model of our/clean-sample attack is more weaker than the perturbation-based attacks. As a result, the most suitable method for comparison is the SOTA clean-sample attack ("Semantic attack"). For the sake of completeness, we also compare our method with the more powerful PBA ("edge-case attack"). Empirically, we find that the DLP outperforms the SOTA clean-sample attack and is comparable with the more powerful edge-case attack in some cases.

**Task I** Due to the low-resolution property of the Fashion-MNIST dataset, one may not obtain fine-grained categories of images other than the original ten classes. Hence, we will compare our proposed method with two SOTA methods on a binary classification task. The new task is to predict if a fashion object is a piece of clothing with label 0 (including 'Top', 'Trouser', 'Pullover', 'Dress', 'Coat', and 'Shirt') or an accessory with label 1 (including 'Sandal', 'Sneaker', 'Bag', ' Boot'). We selected a LetNet classifier whose normal accuracy is 97.2%.

We include the backdoor data preparation process for both "edge-case attack" and "semantic attack" in the appendix. The low-resolution property of the Fashion-MNIST dataset leads the above two SOTA methods to coincide with each other. In the following, we set the backdoor target label to be 0. Then, we tested all the possible categories for backdoor samples and summarized the result in Table. 5. The AccN and AccB of the proposed DLP are 94.1% and 94.7%, respectively. We observe that, for a given AccN of 93%, the proposed DLP obtains the highest AccB against the SOTA methods. Alternatively, for a given AccB of 92%, the AccN of the proposed DLP is slightly lower than the SOTA methods (94.5%). The result is because edge-case attacks modify the training data, which should be more potent than our proposed methods. We follow the same setup above to conduct experiments on **Task II** and **Task III**. We observe similar results as in the Fashion-MNIST, and include the details in the supplement.

Table 5: Test Accuracy (in %) of Task I

| Method | SOTA method | | | | | | Our method |
|---|---|---|---|---|---|---|---|
| Backdoor Category | Top | Trouser | Pullover | Dress | Coat | Shirt | Mixed |
| **AccN** | 94.5 | 93.8 | 93.6 | 91.8 | 90.4 | 92.6 | **94.1** |
| **AccB** | 93.2 | 93.6 | 91.3 | 92.5 | 93.9 | 91.8 | **94.7** |

**Defenses** Because of the weak threat model of our method, many existing defenses against perturbation-based attacks are inappropriate for defending against our attacks. But defenses against label flipping attacks, e.g., label sanitization (Paudice et al., 2018) can be suitably tailored to defend against our method. We found that those defenses are ineffective against our methods. Details are included in the supplementary material.

## 6 CONCLUSION

In this work, we proposed a new type of clean-sample backdoor attack known as DLP. The proposed DLP allows the attacker to choose any number of backdoor samples effectively in terms of test performance. The key ingredient of developing the DLP is a multi-task learning formulation, enabling the attacker to perform well on both normal and backdoor tasks. There are several interesting future problems. One direction is to characterize the optimal backdoor sample theoretically. Finally, on the defense side, it is necessary to create a new defense mechanism to defend the DLP. The supplementary material contains proofs and more experimental studies.

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

# Appendix for
# DLP: Data-Driven Label-Poisoning Backdoor Attack

We include the proof of Theorem 1 and the formal statement of Theorem 2 (and its proof) in Section A. The pseudo-code of algorithms for implementing DLP and the convergence analysis are presented in Section B. Complementary results of the experimental study are included in Section C. Finally, We also present additional experimental results, including investigations on the effect of different sizes of the backdoor sample and results on more complex datasets.

## A  PROOF

### A.1  PROOF OF THEOREM 1

We will rely on the following two lemmas.

**Lemma 1.** (Boucheron et al., 2005) For any $\beta \in \mathbb{R}^d$, we have with probability at least $1 - \delta$,

$$|R_n(\beta) - R(\beta)| \leq 2\mathrm{Rad}_n(H_\beta) + B\sqrt{\frac{\log \frac{1}{\delta}}{2n}}.$$

**Lemma 2.** (Boucheron et al., 2005) For any fixed $W \in \mathbb{R}^p$, we have with probability at least $1 - \delta$,

$$|\widetilde{R}_n(W, \beta) - \widetilde{R}(W, \beta)| \leq 2\mathrm{Rad}_n(G_{W,\beta}) + B\sqrt{\frac{\log \frac{1}{\delta}}{2n}}$$

for any $\beta \in \mathbb{R}^d$.

Back to our main results, we first bound the gap on the normal task.

Invoking Lemma 1, with probability at least $1 - \delta$, we have

$$
\begin{aligned}
R(\widetilde{\beta}) - R(\hat{\beta}) &\leq R_n(\widetilde{\beta}) - R(\hat{\beta}) + 2\mathrm{Rad}_n(H_\beta) + B\sqrt{\frac{\log \frac{1}{\delta}}{2n}}, \\
&= R_n(\widetilde{\beta}) + \frac{n\lambda}{m}\widetilde{R}_n(\widetilde{W}, \widetilde{\beta}) + \frac{\tau}{n}P_m(\widetilde{W}) - \frac{n\lambda}{m}\widetilde{R}_n(\widetilde{W}, \widetilde{\beta}) \\
&\quad - \frac{\tau}{n}P_m(\widetilde{W}) - R(\hat{\beta}) + 2\mathrm{Rad}_n(H_\beta) + B\sqrt{\frac{\log \frac{1}{\delta}}{2n}}, \\
&\leq R_n(\hat{\beta}) + \frac{n\lambda}{m}\widetilde{R}_n(\overline{W}, \hat{\beta}) + \frac{\tau}{n}P_m(\overline{W}) - \frac{n\lambda}{m}\widetilde{R}_n(\widetilde{W}, \widetilde{\beta}) \\
&\quad - \frac{\tau}{n}P_m(\widetilde{W}) - R(\hat{\beta}) + 2\mathrm{Rad}_n(H_\beta) + B\sqrt{\frac{\log \frac{1}{\delta}}{2n}}, \quad (4) \\
&\leq R_n(\hat{\beta}) - R(\hat{\beta}) + \frac{n\lambda}{m}\widetilde{R}_n(\overline{W}, \hat{\beta}) + 2\mathrm{Rad}_n(H_\beta) + B\sqrt{\frac{\log \frac{1}{\delta}}{2n}}, \quad (5)
\end{aligned}
$$

where Eq. (4) holds by definitions of $(\widetilde{W}, \widetilde{\beta})$ (minimizer) and Eq. (5) holds by definitions of $\overline{W}$.

Invoking Lemma 1 on $R_n(\hat{\beta}) - R(\hat{\beta})$ in Eq. (5) with a union bound, with probability greather than $1 - 2\delta$, we have

$$R(\widetilde{\beta}) - R(\hat{\beta}) \leq \frac{n\lambda}{m}\widetilde{R}_n(\overline{W}, \hat{\beta}) + 4\mathrm{Rad}_n(H_\beta) + 2\sqrt{\frac{\log \frac{1}{\delta}}{2n}}. \quad (6)$$

From the definition of $\overline{W}$, there are only $m$ non-negative terms in $\widetilde{R}_n(\overline{W}, \hat{\beta})$ and hence

$$\frac{n\lambda}{m}\widetilde{R}_n(\overline{W}, \hat{\beta}) = \frac{n\lambda}{m}\frac{1}{n}\sum_{i=1}^{n}\mathbf{1}\{y_i \neq \widetilde{y}\}g_{\overline{W}}(x_i)\ell(f_{\hat{\beta}}(x_i), y_i) \leq \frac{n\lambda}{m}\frac{m}{n}B = \lambda B, \quad (7)$$

where $B$ is the uniform upper-bound on the loss function.

Combining Eq. (7) and Eq. (6), the following holds with probability at least $1 - 2\delta$,

$$R(\widetilde{\beta}) - R(\hat{\beta}) \leq \lambda B + 4\mathrm{Rad}_n(H_\beta) + 2B\sqrt{\frac{\log\frac{2}{\delta}}{n}}.$$

Regarding the gap on the backdoor task, we first rewrite

$$\widetilde{R}(\widetilde{W}, \widetilde{\beta}) - \widetilde{R}(\overline{W}, \bar{\beta}) = \widetilde{R}(\widetilde{W}, \widetilde{\beta}) + \frac{m}{n\lambda}R_n(\widetilde{\beta}) + \frac{m\tau}{n^2\lambda}P_m(\widetilde{W})$$
$$- \frac{m\tau}{n^2\lambda}P_m(\widetilde{W}) - \frac{m}{n\lambda}R_n(\widetilde{\beta}) - \widetilde{R}(\overline{W}, \bar{\beta}). \tag{8}$$

Invoking Lemma 2, with probability at least $1 - \delta$, the followings hold

$$\widetilde{R}(\widetilde{W}, \widetilde{\beta}) - \widetilde{R}(\overline{W}, \bar{\beta}) \leq \widetilde{R}_n(\widetilde{W}, \widetilde{\beta}) + \frac{m}{n\lambda}R_n(\widetilde{\beta}) + \frac{m\tau}{n^2\lambda}P_m(\widetilde{W}) - \frac{m\tau}{n^2\lambda}P_m(\widetilde{W}) - \frac{m}{n\lambda}R_n(\widetilde{\beta})$$
$$- \widetilde{R}(\overline{W}, \bar{\beta}) + 2\mathrm{Rad}_n(G_{W,\beta}) + B\sqrt{\frac{\log\frac{1}{\delta}}{2n}},$$

$$\leq \widetilde{R}_n(\overline{W}, \bar{\beta}) + \frac{m}{n\lambda}R_n(\bar{\beta}) + |\frac{m\tau}{n^2\lambda}P_m(\overline{W}) - \frac{m\tau}{n^2\lambda}P_m(\widetilde{W})|$$
$$- \frac{m}{n\lambda}R_n(\widetilde{\beta}) - \widetilde{R}(\overline{W}, \bar{\beta}) + 2\mathrm{Rad}_n(G_{W,\beta}) + B\sqrt{\frac{\log\frac{1}{\delta}}{2n}}, \tag{9}$$

$$\leq \widetilde{R}_n(\overline{W}, \bar{\beta}) + \frac{m}{n\lambda}R_n(\bar{\beta}) + \frac{2m\tau}{\lambda}$$
$$- \widetilde{R}(\overline{W}, \bar{\beta}) + 2\mathrm{Rad}_n(G_{W,\beta}) + B\sqrt{\frac{\log\frac{1}{\delta}}{2n}}, \tag{10}$$

where Eq. (9) holds by the definition of $(\widetilde{W}, \widetilde{\beta})$ (minimizer) and Eq. (10) is because $P_m(\cdot)$ is upper bounded by $2n^2$.

Invoking Lemma 2 on $\widetilde{R}_n(\overline{W}, \bar{\beta}) - \widetilde{R}(\overline{W}, \bar{\beta})$ in Eq. (10) with a union bound, with probability greather than $1 - 2\delta$, we obtain

$$\widetilde{R}(\widetilde{W}, \widetilde{\beta}) - \widetilde{R}(\overline{W}, \bar{\beta}) \leq \frac{2m\tau}{\lambda} + \frac{m}{n\lambda}R_n(\bar{\beta}) + 4\mathrm{Rad}_n(G_{W,\beta}) + 2B\sqrt{\frac{\log\frac{1}{\delta}}{2n}}. \tag{11}$$

For the term $R_n(\bar{\beta})$, we have

$$\frac{m}{n\lambda}R_n(\bar{\beta}) = \frac{m}{n\lambda}\frac{1}{n}\sum_{i=1}^n \ell(f_{\bar{\beta}}(x_i), y_i) \leq \frac{m}{n\lambda}\frac{Bn}{n} = \frac{Bm}{n\lambda}, \tag{12}$$

where $B$ is the uniform upper-bound on the loss function.

Combining Eq. (11) and Eq. (12), the following holds with probability at least $1 - 2\delta$,

$$\widetilde{R}(\widetilde{W}, \widetilde{\beta}) - \widetilde{R}(\overline{W}, \bar{\beta}) \leq \frac{Bm}{n\lambda} + 4\mathrm{Rad}_n(G_{W,\beta}) + 2B\sqrt{\frac{\log\frac{2}{\delta}}{n}} + \frac{2m\tau}{\lambda}.$$

## A.2 FORMAL STATEMENTS OF THEOREM 2

We consider linear classifiers of the form $f_\beta(x) = \beta^\top x$ for $\beta \in \mathbb{R}^d$, with $\|\beta\|_1 \leq W_1$. We assume $\|x_i\|_\infty \leq 1$ for $i = 1, \ldots, n$. For the loss function, following (Boucheron et al., 2005), we take $\ell(f_\beta(X), Y) = \phi(-f_\beta(X)Y)$ where $\phi : \mathbb{R} \to \mathbb{R}_+$. Some common examples are $\phi(x) = \log(1 + e^x)$ for logistic regression and $\phi(x) = \max(0, x)$ for support vector machine.

**Assumption 1.** $\phi(\cdot)$ is uniformly upper bounded by $B$.

**Assumption 2.** $\phi(\cdot)$ is Lipschitz with constant $L$ i.e., $\|\phi(x) - \phi(y)\| \leq L\|x - y\|$.

**Assumption 3.** $t(Z) = g_W(Z)\phi(Z)$ is Lipschitz with constant $L_1$ i.e., $\|t(x) - t(y)\| \leq L_1\|x - y\|$.

**Theorem 3.** (**Linear Case**) Suppose that the assumption 1, 2, and 3 hold. The followings hold with probability at least $1 - 2\delta$:

1. Gap on the normal task:

$$R(\widetilde{\beta}) - R(\hat{\beta}) \leq \lambda B + 4LW_1\sqrt{\frac{\log d}{n}} + 2B\sqrt{\frac{\log \frac{2}{\delta}}{n}},$$

2. Gap on the backdoor task:

$$\widetilde{R}(\widetilde{W}, \widetilde{\beta}) - \widetilde{R}(\overline{W}, \bar{\beta}) \leq \frac{Bm}{n\lambda} + 4L_1W_1\sqrt{\frac{\log d}{n}} + 2B\sqrt{\frac{\log \frac{2}{\delta}}{n}} + \frac{2m\tau}{\lambda}.$$

**Corollary 1.** Under the same assumptions of Theorem 3, by setting $\lambda = \Theta(1/\log n)$, $m = \Theta(\sqrt{n})$ and $\tau = \Theta(1/n)$, two gaps will converge to zero with high probability as $n \to \infty$.

*Proof of Theorem 3 and Corollary 1.* We will be using the following two lemmas.

**Lemma 3.** (Boucheron et al., 2005) For any $\beta \in \mathbb{R}^d$, we have with probability at least $1 - \delta$,

$$|R_n(\beta) - R(\beta)| \leq 2LW_1\sqrt{\frac{\log d}{n}} + B\sqrt{\frac{\log \frac{1}{\delta}}{2n}}.$$

**Lemma 4.** For any fixed $W \in \mathbb{R}^p$, we have with probability at least $(1 - \delta)$,

$$|\widetilde{R}_n(W, \beta) - \widetilde{R}(W, \beta)| \leq 2L_1W_1\sqrt{\frac{2\log d}{n}} + B\sqrt{\frac{\log \frac{1}{\delta}}{2n}}$$

for any $\beta \in \mathbb{R}^d$.

The proof of Lemma 4 is attached at the end of proof of the main theorem.

The proof of Theorem 3 directly follows from Theorem 1 by using explicit forms of the Rademacher Complexity in Lemma 3 and Lemma 4.

Regrading the proof of Corollary 1, it is straightforward to check that every term in the two gaps of Theorem 3 will vanish as $n \to \infty$ with specified hyperparameters.

$\square$

### A.3 PROOF OF LEMMA 4

*Proof.* We will use the following lemma.

**Lemma 5 ((Boucheron et al., 2005)).** Let $\mathcal{F}$ be the class of linear predictors, with the $\ell_1$-norm of the weights bounded by $W_1$. Also assume that with probability one that $\|x\|_\infty \leq X_\infty$. Then

$$\text{Rad}_n(\mathcal{F}) \leq X_\infty W_1\sqrt{\frac{2\log d}{n}},$$

where $d$ is the dimension of data and $n$ is the sample size.

Back to the main proof, recall by definition,

$$\widetilde{R}_n(W, \beta) = \frac{1}{n}\sum_{j=1}^{n} \mathbf{1}\{y_j \neq 1\}g_W(x_j)\phi(\widetilde{y}f_\beta(x_j)).$$

Since $\phi$ is uniformly upper bounder by $B$ and $g_W$ is upper bounded by one, then by Lemma 2,

$$|\widetilde{R}_n(W,\beta) - \widetilde{R}(W,\beta)| \leq 2\text{Rad}_n(G_{W,\beta}) + B\sqrt{\frac{\log\frac{1}{\delta}}{2n}},$$

holds with probability at least $1 - \delta$ where $\text{Rad}_n(G_{W,\beta}) = \mathbb{E}_\sigma[\sup_{\beta\in\mathbb{R}^d} \frac{1}{n}\sum_{j=1}^n \sigma_j \mathbf{1}\{y_j \neq 1\}g_W(x_j)\phi(f_\beta(x_j))]$ with $P(\sigma_i = 1) = P(\sigma_i = -1) = 0.5$ for $i = 1, \ldots, n$.

Without loss of generality, we assume that that there are $s$ samples with ground-truth label 1 with index $n - s + 1, \ldots, n$. Thus,

$$2\text{Rad}_n(G_{W,\beta}) = 2\mathbb{E}_\sigma[\sup_{\beta\in\mathbb{R}^d} \frac{1}{n}\sum_{j=1}^{n-s} \sigma_j g_W(x_j)\phi(f_\beta(x_j))] \tag{13}$$

$$\leq \frac{2L_1}{n}\mathbb{E}_\sigma[\sup_{\beta\in\mathbb{R}^d}\sum_{j=1}^{n-s} \sigma_j f_\beta(x_j)], \tag{14}$$

$$\leq 2L_1 W_1\sqrt{\frac{2\log d}{n}}, \tag{15}$$

where Eq. (14) holds by Lipschitz Composition Principle (Boucheron et al., 2005) and Eq. (15) is by Lemma 5.

$\square$

## B    ALGORITHMS

In this section, we present algorithms for implementing the proposed DLP. We adopt the state-of-the-art techniques in the MTL literature (Sener & Koltun, 2018; Kaiser et al., 2017; Zhou et al., 2017). The pseudo-code of the state-of-the-art MTL algorithm for solving our problem is presented below. The main idea of the algorithm is as follows. We first run gradient descent algorithms on each task in Lines 2-3. Then, to ensure that the overall objective value is decreased, we further update the shared parameter $\beta$ in Lines 4-5. The rationale for obtaining particular $\alpha$ to ensure the decrease of the overall object value can be found in (Sener & Koltun, 2018). Regarding the convergence analysis, under reasonable conditions, the Procedure 1 is shown to find *Pareto stationary points* or local/global optimal points. We refer the interested readers to the references (Sener & Koltun, 2018; Kaiser et al., 2017) for details.

---

**Algorithm 1**

---

**Input:** *Initialization*: Model Parameter $\beta^1$, Selection Parameter $W^1$, Hyperparameters $\lambda, \tau$ and stepsizes $\{\eta_i\}_{i=1}^{T-1}$

---

1: **for** $t = 1, \ldots, T - 1$ **do**
2:     $\beta_t = \beta^t - \eta_t\nabla_\beta L_1(\beta^t)$ // $L_1(\beta) \coloneqq 1/n\sum_{i=1}^n \ell(f_\beta(x_i), y_i)$
3:     $W^{t+1} = W^t - \eta_t\nabla_W L_2(W^t, \beta^t)$ // $L_2(W,\beta) \coloneqq \tau/nP_m(W) + \lambda/m\sum_{j=1}^n \mathbf{1}\{y_j \neq \widetilde{y}\}g_W(x_j)\ell(f_\beta(x_j), \widetilde{y})$
4:     Obtain $\alpha_1^t$ and $\alpha_2^t$ with Procedure 2
5:     $\beta^{t+1} = \beta_t - \eta(\alpha_1^t\nabla_\beta L_1(\beta_t) + \alpha_2^t\nabla_{\beta_t} L_2(W,\beta))$
6: **end for**

---

**Output:** $\beta^T, W^T$

---

## C    COMPLEMENTARY EXPERIMENTAL RESULTS

We provide detailed experimental results that are omitted in the main text due to the page limit in this section.

---

**Algorithm 2** Weight Solver

---

**Input:** *Initialization*: $\boldsymbol{\alpha} = \left(\alpha^1, \alpha^2\right) = \left(\frac{1}{2}, \frac{1}{2}\right)$, $W$ and $\beta$

1: Compute $\mathbf{M}$ st. $\mathbf{M}_{1,2} = \left(\nabla_\beta L_1(\beta)\right)^\top \left(\nabla_\beta L_2(W, \beta)\right)$, $\mathbf{M}_{2,1} = \left(\nabla_\beta L_2(W, \beta)\right)^\top \left(\nabla_\beta L_1(\beta)\right)$
2: Compute $\hat{t} = \arg\min_r \sum_t \alpha^t \mathbf{M}_{rt}$
3: Compute $\hat{\gamma} = \arg\min_\gamma \left((1-\gamma)\boldsymbol{\alpha} + \gamma \boldsymbol{e}_{\hat{t}}\right)^\top \mathbf{M} \left((1-\gamma)\boldsymbol{\alpha} + \gamma \boldsymbol{e}_{\hat{t}}\right)$
4: $\boldsymbol{\alpha}^* = (1-\hat{\gamma})\boldsymbol{\alpha} + \hat{\gamma}\boldsymbol{e}_{\hat{t}}$

---

**Output:** $\boldsymbol{\alpha}^*$

---

## C.1 SELECTED SAMPLES FOR CIFAR10

For task II, we demonstrate the categories of the top $1\%$ selected backdoor training samples in Fig. 4a, and several images of the top-3-category selected training backdoor samples associated with the backdoor label 'Cat', 'Deer' and 'Truck' in Fig. 4b. Similar to the task I in the main text, the top-3-category backdoor samples are semantically consistent with their target labels. For example, the images of 'Deer' and 'Dog' resemble the images of 'Horse' most than any other categories in the Fashion-MNIST dataset. Such observations provide concrete evidence to support the conjecture that backdoor images should be similar to their backdoor label category (Bagdasaryan et al., 2020). The similarity is measured in terms of semantic meanings in our case.

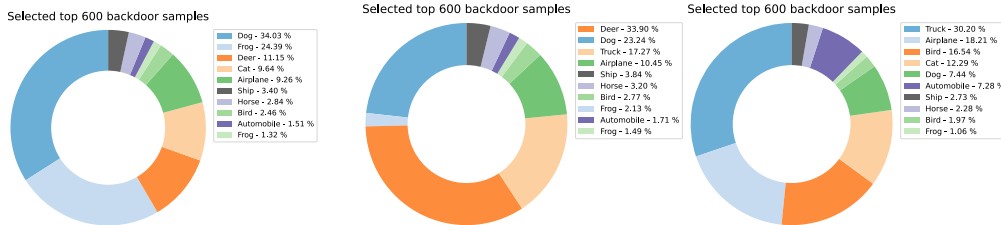

(a) Pie-charts of selected backdoor samples corresponding to backdoor label Cat (left), Horse (middle) and Automobile (right).

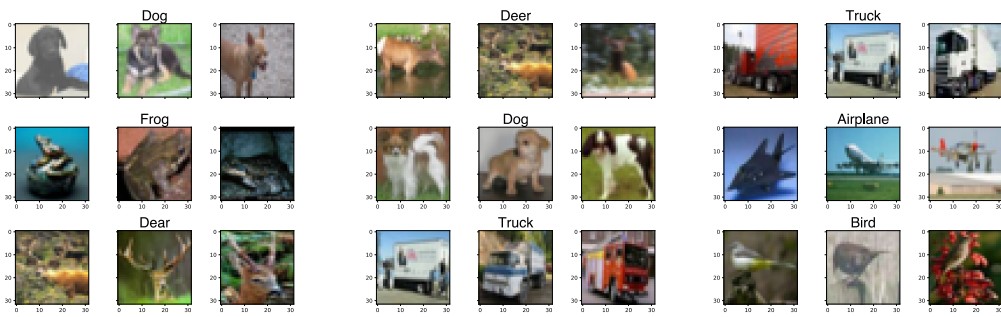

(b) Snapshots of top-3-category selected backdoor samples corresponding to backdoor label Cat (left), Horse (middle) and Automobile (right).

Figure 4: Illustration of (a) selected categories of backdoor samples in pie chart and (b) selected examples for CIFAR10 dataset.

## C.2 Test Performances on GTSRB

In this section, we test the proposed method on the GTSRTB dataset. There are in total 43 types of traffic signs (labels) in GTSRB, and many of them are of the same type, e.g., "20 speed","30 speed". For the sake of clarity, we only select one of each types for testing.

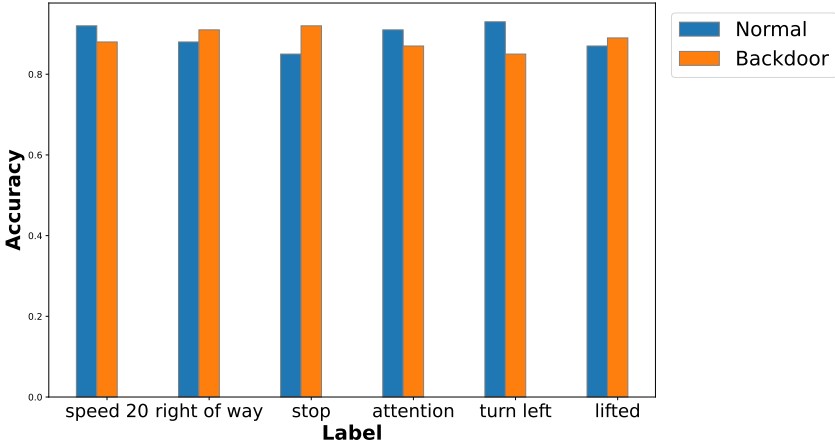

Figure 5: Accuracy of proposed method on GTSRB

## C.3 Comparison with state-of-the-art on CIFAR10

For semantic backdoor attacks, following the framework in (Bagdasaryan et al., 2020), we relabel a whole category of the training data with the same semantic meaning. For example, we can relabel images of Top (with ground-truth label 1) with label 0. For edge-case attacks, we follow the ideas in (Wang et al., 2020) to first create a new training dataset by excluding the samples of one whole category. Then, we relabel the samples of the previously excluded sub-category and added them to the previous training data to form a new training data.

We follow the same setup for **Task II** in the main text. The two SOTA methods to be compared are listed below. For semantic attacks, the work of (Bagdasaryan et al., 2020) relabeled a whole category of the training data. For edge-case attacks, the authors in (Wang et al., 2020) first relabeled the images of Southwest Airplanes (NOT in CIFAR10) as Truck and then injected the relabeled sample-label pairs into the training data. For completeness, we also test for different backdoor labels as summarized in Table 6. The proposed DLP consistently outperforms the SOTA clean-sample attack for all backdoor target labels. Also, the proposed DLP is comparable with the more powerful edge-case attack in some cases, e.g., a backdoor label of 'Dog'.

Table 6: Test Accuracy (in **%**) on CIFAR10

| Method | DLP | | Semantic | | Edge-case | |
|---|---|---|---|---|---|---|
| Accuracy | AccN | AccB | AccN | AccB | AccN | AccB |
| Label: Frog | 84.2 | 85.7 | 83.2 | 81.6 | 85.1 | 84.8 |
| Label: Truck | 86.9 | 87.2 | 79.3 | 82.9 | 86.5 | 89.2 |
| Label: Dog | 83.5 | 89.3 | 82.1 | 87.3 | 88.5 | 90.9 |
| Label: Horse | 84.1 | 89.2 | 83.2 | 81.5 | 87.5 | 91.1 |

## C.4 Experimental results under defense mechanisms

In this section, we test the proposed method under certain defenses. As mentioned in the main text, because of the weak threat model of (centralized) clean-sample backdoor attacks, many existing defenses against perturbation-based attacks are inappropriate for defending against our attacks.

But defenses against label flipping attacks, e.g., label sanitization (Paudice et al., 2018) and label ceritication (Rosenfeld et al., 2020) can be suitably tailored to defend against our method. The results are summarized in Fig. 6 and Fig. 7 respectively. It can be concluded from the figure that the proposed method can escape the two defenses.

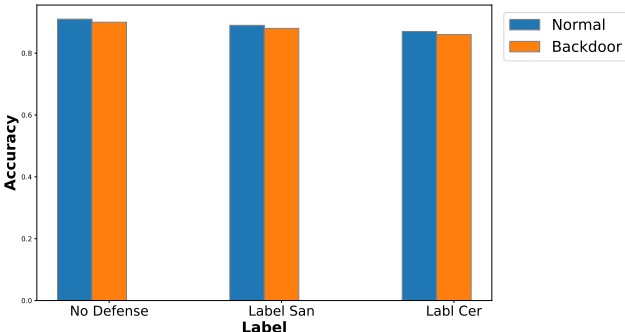

Figure 6: Test performance on Fashion-MNIST

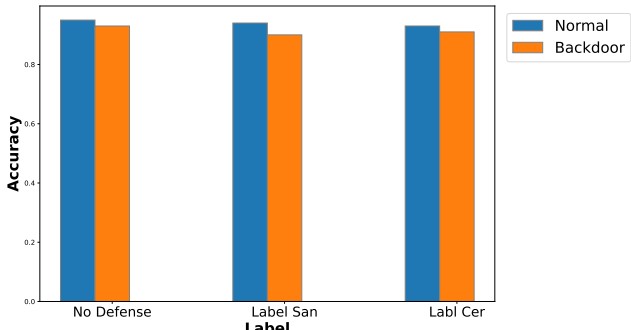

Figure 7: Test performance on CIFAR10

