# OpenReview forum: "DLP: Data-Driven Label-Poisoning Backdoor Attack"
_ICLR.cc/2023/Conference — Submitted to ICLR 2023_

### Official Review · Reviewer_e5gT · 2022-10-24

**Confidence:** 4
**Correctness:** 3
**Technical Novelty And Significance:** 3
**Empirical Novelty And Significance:** 2
**Recommendation:** 3

**Clarity, Quality, Novelty And Reproducibility:**

Detailed comments are given in the **Strength And Weaknesses**. Below is a summary:

#### **Clarity**: the paper needs some work to explain the proposed approach in an accessible language. For further info, please see the first part of the weaknesses.

#### **Quality**: as mentioned above, the paper requires more straightforward language, better motivation, and further experimentation. These issues place the paper in a moderate quality.

#### **Novelty And Reproducibility**: The paper presents an interesting attack in terms of novelty. However, different aspects of the current work need to be investigated better. Furthermore, in terms of reproducibility, the paper neither provides the code nor a detailed set of hyper-parameters for the results to be re-created.

**Strength And Weaknesses:**

### Strengths:
- The proposed method seems interesting. In particular, it is intriguing to see that one can find semantically similar images to the target class and use those samples as the backdoor activation.

- The theoretical analysis seems relevant to the paper.

- The experimental results, especially Figures 3 and 4, beautifully depict the overall idea of the paper.

### Weaknesses:
- A major concern with the current version is the presentation of the paper, especially the proposed method. Many aspects of the proposed method are left to the reader's interpretation, and the paper does not discuss them properly. These include:
   1. In Eq. (1), for the first time the reader faces the objective function for training the scoring mechanism $g_{W}(\boldsymbol{x}_i)$. However, the paper does not properly discuss the objective function's meaning. In particular, what does Eq. (1) imply beyond mathematical terms? Please explain this in accessible language to the readers. Additionally, the symbol $m$ is used in this equation for the first time. However, its definition is left toward the end of Eq. (2) in the next page.

  2. In the definition of $P_{m}(W)$, it takes the reader a while to figure out the equivalency of the minimizer of $P_{m}(W)$ with the constraints in Eq. (2). The paper should mention that  $P_{m}(W)$ is always non-negative, and why is that the case? Discuss how the second term $\sum_{j=1}^n g_W\left(\boldsymbol{x}_j\right)\left(1-g_W\left(\boldsymbol{x}_j\right)\right)$ is always non-negative and remind the reader that you assumed $0 \leq g_W\left(\boldsymbol{x}_j\right) \leq 1$.

  3. A similar pattern can be seen during the theoretical results in Sec. 4.2. While the theory is mathematically presented, it would be better to accompany it with further discussions to let the readers grasp what those theorems mean in simplistic language. Also, could you please elaborate on why Assumption 1 is practical?

- Another critical issue with the current work is its motivations. In Sec. 3.1. it is mentioned that the proposed clean-sample threat model is **weaker** in terms of its requirements. However, the current attack requires being present during training, which makes it easier for the attacker. The previous backdoor poisonings could be used to poison the data only without requiring any control over the training mechanism. As such, the requirements of the current attack are not weaker than existing methods. Besides, there is a great body of literature on *feature collision* attacks that share many characteristics with the current approach, e.g., see [1,2]. However, none of them are mentioned and discussed in the paper.

- Finally, the experimental results need more investigations and baselines. The current method is indeed inherently different from perturbation-based backdoor attacks, but comparing the proposed method against them would help the readers understand the bigger picture. More importantly, a comparison on existing baselines for the detection of backdoor samples could show how much the selected backdoor samples in the paper are successful in circumventing the existing defense mechanisms. Also, investigating the transferability of the backdoor samples between models is another important venue. In other words, what does happen if one generates a poisoned dataset for a model $f_{1}(\cdot)$ but trains a classifier $f_{2}(\cdot)$ with that data? Could the backdoors be transferred between models? Additionally, what are the proposed method's computational burdens compared to vanilla training? Finally, a detailed investigation of the effects of different hyper-parameters on the model's performance over tasks I-III is also missing.

**Minor comment**: The remark mentioned in footnote 1 is inaccurate. One could poison the training data with perturbation-based triggers and maintain the training data size. For instance, see [3].


[1] Shafahi, Ali, et al. "Poison frogs! targeted clean-label poisoning attacks on neural networks," *NeurIPS*, 2018.

[2] Saha, Aniruddha, et al. "Hidden trigger backdoor attacks," *AAAI*, 2020.

[3] Turner, Alexander, et al. "Label-consistent backdoor attacks," *arXiv preprint arXiv:1912.02771* (2019).

**Summary Of The Paper:**

This paper proposes a new backdoor attack. Under the proposed threat model, the attacker aims to use semantically similar images to the target class to activate the backdoor. For instance, green cars can be used as semantically similar images to frogs in CIFAR-10. As such, one may re-label those green car images into frogs to create a poisoned data set. The paper argues that under this regime, known as clean-sample backdoor attacks, the attacker requires a weaker set of requirements and can only activate the backdoor without modification of the underlying test samples. This contrasts perturbation-based backdoor attacks that need to attach triggers during test time. To find semantically similar images for a particular target class, the paper uses a score function over the entire set of data that assigns a value between 0 and 1 to any specific input. This score function is supposed to indicate which samples (that come from a non-target class) have similar features to the target class so that they can be used as backdoors. This function is trained in conjunction with the target neural network and is used to select a handful of samples as the backdoor data. Theoretically, the paper shows that as the number of training samples grows, the generalization bounds between the optimal solutions for benign and backdoor empirical risks approach zero. Experimental results over Fashion-MNIST, CIFAR-10, and GTSRB are provided.

**Summary Of The Review:**

Based on my detailed comments above, I feel the paper needs some work. As such, I vote for its rejection at this stage. Meanwhile, I need to read my peers' reviews and wait for the authors' responses to make my final decision.

---

### Official Review · Reviewer_9Uiq · 2022-10-24

**Confidence:** 4
**Correctness:** 3
**Technical Novelty And Significance:** 3
**Empirical Novelty And Significance:** 2
**Recommendation:** 5

**Clarity, Quality, Novelty And Reproducibility:**

The method description is clear and the theoretical analysis is sufficient. In the experimental part, the validity analysis of the results is relatively redundant. The method comparison and defense result analysis can be elaborated more.

**Strength And Weaknesses:**

Strength

Compared with the current mainstream backdoor attack methods, this method has lower requirements and stronger concealment. The theoretical derivation of the backdoor scoring mechanism is very detailed and the method is convincing.

Weaknesses
1. The dataset used in the experiment is too simple, and there is no experiment on the complex dataset. Compared with SOTA, the improvement is not obvious, and the experimental persuasion needs to be improved.
2. I doubt the assumption of the attacker settings, e.g., “it is difficult for attackers to access users’ input queries in the test phase”. Actually, the ML service is often public and the attacker could be the user too, such as the pre-trained models deployed on the cloud.
3. Another question is about the claim, “clean-sample backdoor attacks are more malicious than perturbation-based attacks since they do not modify features of input data.” From the empirical view, the clean-sample backdoor attacks always incur more performance loss at the clean test set while the perturbation-based method seldom hurt the performance of the model on other clean inputs. Maybe more evidence should be involved to support this claim.

**Summary Of The Paper:**

The author proposes a new backdoor attack-DLP, which only needs to modify the training set labels to threaten the model. And this backdoor attack method uses a data-driven backdoor scoring mechanism, which can take effect within any backdoor sample size. Finally, the effectiveness of the method is proved by theory and experiment.

**Summary Of The Review:**

The author proposes a backdoor attack method named DLP, which uses a data-driven backdoor scoring mechanism to select data in a novel and fully theoretically derived way. However, the experiment is slightly insufficient, and further experiments such as on ImageNet and the corresponding baseline need to be added to prove the advantages of this method.

---

### Official Review · Reviewer_xGur · 2022-10-25

**Confidence:** 4
**Correctness:** 3
**Technical Novelty And Significance:** 2
**Empirical Novelty And Significance:** 2
**Recommendation:** 5

**Clarity, Quality, Novelty And Reproducibility:**

It is well written, and easy to follow.
My main concern is the lack of novelty, and the proposed framework has a strong limitation, as mentioned above (i.e., its performance heavily depends on the training dataset.)

**Strength And Weaknesses:**

positives:
+ clean-sample backdoor attack is an interesting backdoor attack, which is pretty stealthy. Thus it needs to be explored in advance.
+ theoretically analysis is given to justify the propose algorithm could leads to a proper clean-sample attack.

negatives:
- the proposed method is straightforward, which just introducing a term to learn the scoring function, but the formulation of scoring function g_w is less discussed. It could be an neural network or even a linear function. More important, the proposed method would heavily depends on the training data, since it just select samples from them. As a result, it is hard to control the final performance if the training dataset is not easily to be backdoored. I suggest to introduce some new samples into the dataset which could efficiently improve the performance of the attack.



**Summary Of The Paper:**

This paper focus on the clean-sample backdoor attack, which does not modify images but just flips image labels. One key step of  clean-sample attack is how to select samples/features to conduct label poisoning. This paper proposes a scoring mechanism to select proper samples, which allows to effectively select samples for arbitrary backdoor sample size.

**Summary Of The Review:**

As mentioned above, the proposed method need to be significantly improved before being accepted.

---

### Official Review · Reviewer_1LBS · 2022-10-27

**Confidence:** 4
**Correctness:** 2
**Technical Novelty And Significance:** 3
**Empirical Novelty And Significance:** 2
**Recommendation:** 3

**Clarity, Quality, Novelty And Reproducibility:**

The paper is clearly written. The proposed idea is sufficiently novel and should be reproducible if the code is shared.

**Strength And Weaknesses:**

The paper is well written and clearly presented. The experimental results show the effectiveness of the method and the visual examples provide good illustrative examples.

My concerns are mainly regarding the premise of the proposed method/problem and insufficient experiments.

In my opinion, during backdoor attacks, creating out-of-distribution data (whether through subtle manipulation of the image, like WaNet, or explicit insertion of triggers) at the test time is necessary. The attacked model should behave normally at any in-distribution data, as users expected. If a model always misclassifies "green cars" as "frogs", it defeats the purpose of a stealthy attack. This makes it hard to be convinced the overall approach of the paper is a valid attacking scenario.

Furthermore, although the experiments are illustrative of the proposed idea. They are far from sufficient. Currently, only one clean-sample attack is compared with. Considering many attacking and defending algorithms have been developed in recent years, the method should be compared with these attacking methods with different poisoning rate. Also the method should be evaluated against defending/detection algorithms to show it is stealthy enough. It would also be interesting to see how the method performs using adversarial training.



**Summary Of The Paper:**

This paper proposes a new attack method. Unlike previous methods that perturbs the training data, the proposed method only perturb the label of a selected subset of training data. This essentially finds a small population of data and train the model to misclassify them consistently. The benefit is that at inference stage, the attacker does not need to manipulate test images. The model will automatically misclassify the selected population. Experimental results show that the method can train models with satisfying clean data accuracy and misclassification rate, even with a small poisoning rate.

**Summary Of The Review:**

The paper is well written and the idea is novel. But the premise of the solution is not convincing, and experiments are very limited.

---

### Decision · Program_Chairs · 2023-01-20

**Decision:**

Reject

**Justification For Why Not Higher Score:**

Explained in part 1.

**Justification For Why Not Lower Score:**

N/A

**Metareview: Summary, Strengths And Weaknesses:**

The paper introduces a new type of clean-sample backdoor attack, namely DLP backdoor attack, allowing attackers to backdoor effectively. While previous methods perturb the training data, DLP only perturbs the label of a selected subset of training data. This alleviates the need to manipulate images at test time. One main concern of the reviewers, which is agree with, is that if the attack changes the label of in-distribution examples without modifying them, it becomes easy to detect at test time. Testing the proposed methods against existing backdoor attack detection methods would significantly strengthen the claims of the paper.